# Impact of COVID-19 on Patients with Decompensated Liver Cirrhosis

**DOI:** 10.3390/diagnostics13040600

**Published:** 2023-02-06

**Authors:** Tudor Voicu Moga, Camelia Foncea, Renata Bende, Alina Popescu, Adrian Burdan, Darius Heredea, Mirela Danilă, Bogdan Miutescu, Iulia Ratiu, Teofana Otilia Bizerea-Moga, Ioan Sporea, Roxana Sirli

**Affiliations:** 1Department of Gastroenterology and Hepatology, “Victor Babeș” University of Medicine and Pharmacy, Piața Eftimie Murgu 2, 300041 Timișoara, Romania; 2Center of Advanced Research in Gastroenterology and Hepatology, “Victor Babeș” University of Medicine and Pharmacy, 300041 Timisoara, Romania; 3Department of Pediatrics-1st Pediatric Discipline, “Victor Babeș” University of Medicine and Pharmacy, 300041 Timisoara, Romania

**Keywords:** COVID-19, decompensated liver cirrhosis, outcome

## Abstract

The aim of this study was to assess the impact of COVID-19 infection on patients with decompensated liver cirrhosis (DLC) in terms of acute-on-chronic liver failure (ACLF), chronic liver failure acute decompensation (CLIF-AD), hospitalization, and mortality. In this retrospective study, we analyzed patients with known DLC who were admitted to the Gastroenterology Department with COVID-19. Clinical and biochemical data were obtained to compare the development of ACLF, CLIF-AD, days of hospitalization, and the presence of independent factors of mortality in comparison with a non-COVID-19 DLC group. All patients enrolled were not vaccinated for SARS-CoV-2. Variables used in statistical analyses were obtained at the time of hospital admission. A total of 145 subjects with previously diagnosed liver cirrhosis were included; 45/145 (31%) of the subjects were confirmed with COVID-19, among which 45% had pulmonary injury. The length of hospital stay (days) was significantly longer in patients with pulmonary injury compared to those without (*p* = 0.0159). In the group of patients with COVID-19 infection, the proportion of associated infections was significantly higher (*p* = 0.0041). Additionally, the mortality was 46.7% in comparison with only 15% in the non-COVID-19 group (*p* = 0.0001). Pulmonary injury was associated with death during admission in multivariate analysis in both the ACLF (*p* < 0.0001) and the non-ACLF (*p* = 0.0017) group. COVID-19 significantly influenced disease progression in patients with DLC in terms of associated infections, hospitalization length, and mortality.

## 1. Introduction

It is well known that bacterial infection and systemic inflammation play an essential role in dictating the course of liver cirrhosis (LC) by aggravating decompensation and organ dysfunction [1].

One large observational study was performed to better describe and diagnose decompensation in LC. The CANONIC study established diagnostic criteria for acute on chronic liver failure (ACLF) and introduced the concept of acute decompensation (AD) [2]. AD is defined as acute development of one or more of the following: first episode or recurrent ascites grade 2–3 within 2 weeks, first episode of hepatic encephalopathy in patients with a previously normal neurological state, acute episode of gastrointestinal bleeding, and bacterial infections. AD is based on acute development of these complications which define the decompensation state of LC. The CANONIC study introduced bacterial infections as AD because of the high prevalence among patients with LC and because they worsen prognosis in these patients [3].

ACLF was defined by the CANONIC study as the development of failure in different organs: liver, brain, kidney, coagulation, and circulation. The number of organ failures subclassifies ACLF into grades 1, 2, or 3 depending on the association with one, two, or three or more organ failures.

AD was confirmed as a distinct entity from ACLF and by a specific score (CLIF-C AD). The 3 month mortality can be more accurately predicted and differs from the traditional Child–Pugh score or MELD score [4]. At admission, in all patients with LC, the CLIF-C OF score for ACLF should be applied in order to allow the correct classification into a group with ACLF or AD.

Decompensated liver cirrhosis (DLC) is a systemic disease, with multiorgan/system dysfunction. DLC patients are prone to infection due to their altered immune system [5]; thus, the management of DLC primarily involves the treatment of pathogenic factors [6].

Even though the SARS-CoV-2 virus is known to primarily affect the lungs, liver injury can appear in up to half of the infected patients [7]; hence, elevated transaminases can occur. The severity and progression of COVID-19 seem to indicate the degree of liver injury [8], which might be linked to angiotensin-converting enzyme 2 (ACE2) receptors that are also found in the liver (more exactly at the bile duct level); hence, severe inflammation known as the “cytokine storm” is amenable for liver injury [9,10,11].

The aim of the study was to assess the influence of COVID-19 infection on patients with previously known DLC in terms of ACLF, CLIF-C AD, development of complications, hospitalization, and mortality in a tertiary gastroenterology department.

## 2. Materials and Methods

### 2.1. Study Design and Participants

A retrospective study was performed on patients with LC and SARS-CoV-2 superimposed and cirrhosis alone, admitted to a tertiary department of gastroenterology. Data were collected between 17 November 2020 and 28 November 2021. All cases with laboratory-confirmed SARS-CoV-2 infection by reverse transcription PCR (RT-PCR) on nasopharyngeal swabs in patients with LC >18 years old, with any symptoms and any severity grade, were included in the analysis. None of the patients included in the study was vaccinated against SARS-CoV-2 infection. Exclusion criteria were age below 18, pregnancy, oncologic patients, surgical emergencies, and liver failure without underlying LC.

Data were collected on the basis of history (including previous episodes of AD or ACLF), as well as clinical and paraclinical examination: laboratory tests, imagistic findings, and events that may be potential precipitating factors of both AD and ACLF, such as active alcoholism (more than 14 drinks per week in women and more than 21 drinks per week in men within the previous 3 months), bacterial infection, gastrointestinal hemorrhage, therapeutic paracentesis, or other new events which could impair liver function within 2 weeks before presentation.

All data were collected retrospectively. The presence of AD and the development of ACLF, as well as their influence on mortality and outcome in patients with SARS-CoV-2 infection, was assessed, as shown in the flowchart (Figure 1).

### 2.2. Variables 

Liver disease was reported by the clinician. All included patients presented LC which was categorized on the basis of the Child–Pugh score and MELD score (the model for end-stage liver disease included serum sodium, creatinine, bilirubin, and international normalized ratio). The diagnosis of cirrhosis was based on previous clinical signs and findings provided by laboratory test results, endoscopy, and/or radiologic imaging. 

DLC is defined as the presence or history of overt ascites (or pleural effusion with increased serum ascites albumin gradient (>1.1 g/dL)), overt hepatic encephalopathy (West Haven grade > 2) and variceal bleeding [12].

AD was defined as any first or recurrent grade 2 or 3 ascites within less than 2 weeks, first or recurrent acute hepatic encephalopathy in patients with previous normal consciousness, acute gastrointestinal bleeding, and any type of acute bacterial infection [13].

ACLF was defined according to the EASL-CLIF acute-on-chronic liver failure in cirrhosis (CANONIC) study [2]. ACLF grade 1 was defined as single kidney failure or renal dysfunction (serum creatinine 1.5–1.9 mg/dL) plus nonrenal organ failure and/or brain dysfunction with grade 1–2 hepatic encephalopathy. ACLF grades 2 and 3 was defined by the presence of two or three organ failures. Organ failures are defined by the CLIF sequential organ failure assessment (CLIF-OF) as follows: kidney failure, serum creatinine ≥ 2 mg/dL or renal replacement; liver failure, bilirubin ≥ 12 mg/dL; coagulation failure, INR ≥ 2.5 and/or platelet count 20 × 10^9^/L; cerebral failure, grade 3–4 hepatic encephalopathy; respiratory failure, a ratio of partial pressure of arterial oxygen to fraction of inspired oxygen (FiO_2_) of 200 or a ratio of pulse oximetric saturation (SpO_2_) to FiO_2_ of 200; circulatory failure, need for vasopressor agents to maintain arterial pressure ≥90 mmHg.

### 2.3. Ethical Approval

Because this was an observational study, performed with a retrospective design using a database and medical records, informed consent was waived. However, all patients signed an informed consent in written form before hospitalization.

### 2.4. Statistical Analysis

The statistical analysis was performed using MedCalc Version 19.4 (MedCalc Software Corp., Brunswick, ME, USA) and Microsoft Office Excel 2019 (Microsoft for Windows). Descriptive statistics were used for demographic, anthropometric, and clinical data. The distribution of numerical variables was tested using the Kolmogorov–Smirnov test, and results were summarized as the median and interquartile range (IQR) or mean and standard deviation, as appropriate; categorical variables were expressed as frequencies and percentages. Student’s t-test was used for group comparisons of continuous variables with a normal distribution, and a Mann–Whitney U-test was applied for variables with a non-normal distribution. Group comparisons of categorical variables were performed using Pearson’s χ^2^ test. A *p*-value < 0.05 was considered significant for each statistical test. Cox regression analysis was used to identify factors associated with 30 day mortality.

## 3. Results

### 3.1. Demographic and Clinical Characteristics of the Patients

A total of 145 subjects with previously diagnosed liver cirrhosis were included; 45/145 (31%) of the subjects were confirmed with SARS-CoV-2 between 17 November 2020 and 28 November 2021. The demographic and clinical characteristics of the patients are summarized in Table 1.

### 3.2. COVID-19 Assessment

Patients were divided into two subgroups according to the presence and severity of pulmonary injury determined by computed tomography at the time of admission.

The subgroup of patients with pulmonary injury included 20 individuals, 65% of whom were males with a median age of 63 years (37–78). The extent of their pulmonary injury ranged from 5% to 85%.

The length of hospital stay (days) was significantly longer in patients with pulmonary injury compared to those without (*p* = 0.0159). No differences were found regarding liver enzyme levels and inflammatory syndrome marker levels in the two groups of subjects (Table 2).

### 3.3. Impact of COVID-19 on Patients with LC

When comparing subjects with COVID-19 infection with those without, the length of hospital stay (days) was significantly longer for COVID-19 subjects, 10 (1–34) vs. 5 (1–23) (*p* < 0.0001). No differences were found between liver enzymes levels in the two groups; the median AST value was 68 IU/L (19–1293) for subjects with COVID-19 infection and 63.5 IU/L (17–3529) for those without (*p* = 0.5011), while the median ALT value was 34 IU/L (10–354) for subjects with COVID-19 and 37 IU/L (9–1669) for those without (*p* = 0.6395). In the group of patients with COVID-19 infection, the proportion of associated infections was significantly higher compared to those without (*p* = 0.0041); instead, no differences were found regarding the other factors of acute decompensation: ascites, spontaneous bacterial peritonitis, encephalopathy, and upper-gastrointestinal bleeding (Figure 2). A more detailed analysis of the associated infections occurring in the two groups is presented in Figure 3.

### 3.4. Overall and COVID-19-Specific Mortality

In the group of subjects with liver cirrhosis and COVID-19 infection, 48.9% (22/45) had ACLF, whereas, in the subjects without COVID-19 infection 30% (30/100) had ACLF (Table 3). CLIF C-ACLF score was determined for all the subjects with ACLF, and CLIF-C AD score was determined for subjects without ACLF. A more detailed comparative analysis between subjects with COVID-19 and those without is summarized in Table 3. 

The overall mortality during admission was 24.8% for our study group. In the COVID-19 group, the mortality was 46.7% in comparison with only 15% in the non- COVID-19 group (*p* = 0.0001) (Table 3).

Cox regression analysis was used to identify factors associated with death during admission. In univariate regression analysis, the following parameters were associated with death during admission: pulmonary injury (*p* < 0.0001), the presence of COVID-19 infection (*p* < 0.001) or other associated infection (*p* < 0.001), Child–Pugh score (*p* = 0.012), MELD score (*p* = 0.004), ACLF grade (*p* = 0.017), CLIF-OF score (*p* < 0.001), CLIF C-ACLF (*p* = 0.003), and CLIF C-AD score (*p* < 0.001).

In multivariate analysis, two models were built, one for subjects with ACLF and one for subjects without. For subjects with ACLF (*n* = 52), the model including pulmonary injury (*p* < 0.0001), Child–Pugh score (*p* = 0.012), MELD score (*p* = 0.004), and CLIF C-ACLF score (*p* = 0.003) was associated with death during admission. In subjects without ACLF (*n* = 93), the best model for predicting death during admission included the following parameters: pulmonary injury (*p* = 0.0017) and CLIF C-AD score (*p* = 0.0004). The predictors of mortality are summarized in Table 4.

## 4. Discussions

It is known that the most lethal complication of LC is the development of ACLF with subsequent hepatic and extrahepatic organ failure. Considering the high prevalence among both LC patients (30%) and outpatients (25%) and the poor outcome with a high mortality rate (50%), it is important to identify new predictors that can distinguish high-risk patients [2,6,14,15]. According to Piano et al. [14], low mean arterial pressure, ascites, anemia, and MELD score were associated with the risk of developing ACLF in outpatients with advance liver disease. AD of patients with LC is the main ingredient for the development of ACLF. CLIF-C AD score was proposed to evaluate AD as a diagnostic feature for ACLF [4,6]. Excessive systemic inflammation is the background of AD and, thus, ACLF that leads to organ failure and high 28-day mortality [4,6,16].

Our study showed that SARS-CoV-2 infection in patients with DLC influences the development of organ failure, associated infection, and mortality. Given the data showing that associated infections were more frequent in the SARS-CoV-2 group, we might conclude that patients with LC and COVID-19 superimposed might be prone to bacterial infections and, thus, to poor outcome. 

Comorbidities are frequently cited as risk factors for severe forms of COVID-19. Hypertension, chronic kidney disease, diabetes mellitus, and cancer have been independently associated with mortality in COVID-19 patients, with chronic kidney disease being the most common comorbidity according to a meta-analysis that included 375,859 patients [17]. 

Patients with LC admitted in our center during a 12 month period were evaluated; among them, 45/145 were diagnosed with SARS-CoV-2 infection at or during admission. According to the World Health Organization SARS-CoV-2 tracking variants and National Ministry of Health, 36 of our patients were infected with the Alpha variant (B.1.1.7) and 13 were infected with the Delta variant (B.1.617.2) [18]. We have no clear data regarding community-acquired or healthcare-acquired SARS-CoV-2 infection mainly because of increased need of medical care and hospitalization owing to medical assistance and frequent complication of LC. The patients received treatment in agreement with the national recommendations [18,19,20]. Some of the antiviral treatments or antibiotics used in COVID-19 patients can influence gastrointestinal symptoms such as diarrhea, nausea, or elevated liver enzymes [21,22,23].

In our cohort, favipiravir was the most prescribed drug among other supportive treatment (cortisol, heparin, vitamin C, vitamin D3, antacid H2 blocker, and zinc). In our study, there were no differences between the two subgroups (with and without pulmonary injury) of patients with COVID-19 infection from the point of view of laboratory findings, which is in accordance with the study of Wang [24], where the adverse events of patients receiving antiviral treatment were not statistically different from those of the placebo group.

Since there were no side-effects associated with the treatment, all patients who were selected for treatment completed the cure successfully.

According to our findings, pulmonary injury depicted on computed tomography in the SARS-CoV-2 group did not influence CRP, liver enzymes, or even hospitalization. The authors of [25] highlighted the potential for liver injury related to antiviral treatment, as well as secondary to the impact of the SARS-CoV-2 virus on ACE2 receptors in the biliary ducts. However, among our COVID-19 group with lung injury as shown on CT scans, none of the abovementioned hypotheses were observed.

The complexity of the RAS (renin–angiotensin system) and its modulation in liver injury are further illuminated by the overexpression of ACE2 in rat and cirrhotic human liver. ACE2 is a crucial component in the propagation of the SARS coronavirus and is a functional receptor for it. Patients with SARS frequently have liver damage, and new research has shown that the SARS coronavirus induces a unique form of hepatitis that is characterized by lobular inflammation and hepatocyte death. An explanation for the discovery that SARS infects the liver may be found in the study’s conclusion that the principal cellular receptor for the disease is present in the liver and increases in liver injury [26]. In another multicentric study, which enrolled 104 patients with SARS-CoV-2 infection, 81 patients with digestive symptoms were more likely to suffer liver injury because of the upregulation of ACE-2 expression in the liver tissue [27].

The study of Solopov et al. [28] focused on the role of alcohol intake and SARS-CoV-2 infection, and demonstrated that, in mice treated with alcohol, a histologic analysis of the lung tissue revealed aberrant immune cell recruitment in the alveolar space, abnormal parenchymal architecture, and worsened Ashcroft scores. Lung tissue homogenates from mice on an alcohol diet demonstrated upregulation of ACE2, in addition to the activation of proinflammatory biomarkers compared with mice on a control diet. This article could be very important in future research on alcoholic-induced liver disease and SARS-CoV-2 infection, helping in the development of therapeutic strategies.

In comparison to the study of Chen et al. [29], which reported a 50% ALAT increase in a cohort of 202 patients with COVID-19 (38% of which had nonalcoholic fatty liver disease), and the study of Iavarone et al. [30], which first described the impact of COVID-19 on patients with liver cirrhosis and noted that acute liver injury was present in half of the SARS-CoV-2-infected patients with previously normal transaminase, our study, despite comparing two distinct groups of patients with LC, one superimposed with COVID-19, did not show any significant differences between the two groups from the point of view of Child–Pugh score, MELD score, and liver enzymes. Furthermore, when compared within the COVID-19 group, i.e., those with and without lung injury, there were no significant differences between the subgroups in terms of liver enzymes (Table 2).

One explanation might be the differences in the mean age of the subjects (67 vs. 62.2 years) between studies, which could explain a frail and comorbid cohort of patients in Iavarone et al.’s study in comparison to ours. The number of patients with ACLF was also higher in our study [30] (22 vs. 13); however, when comparing the group of patients with the same baseline characteristics, the difference was not significant. Nonetheless, in both studies, CLIF-C, CLIF-OF, and MELD scores and lung injury were associated with mortality, which was significantly higher in COVID-19 group. The fact that COVID-19 superimposed on patients with LC significantly influenced organ failure, associated infections, hospitalization, and mortality was, by some means, anticipated. However, a curious result is shown in Table 2, where the impact of lung injury did not influence laboratory findings in the COVID-19 group, for which we could find no explanation.

There were also some limitations to this study worth considering, such as its retrospective and monocentric nature, the small cohort of patients with LC and COVID-19, and the exclusion of the ICU patients due to the limited access. Group heterogeneity (ALD as the main etiology of LC) might also have been a limitation. Unfortunately, there are no data regarding comorbidities that might have influenced the mortality rate; however, if the patients were admitted to the gastroenterology department, even though they might have had other comorbidities, the main issues were related to GI pathology.

## 5. Conclusions

COVID-19 predisposed patients with decompensated liver cirrhosis to increased risk of associated infections, organ failure, and mortality.

## Figures and Tables

**Figure 1 diagnostics-13-00600-f001:**
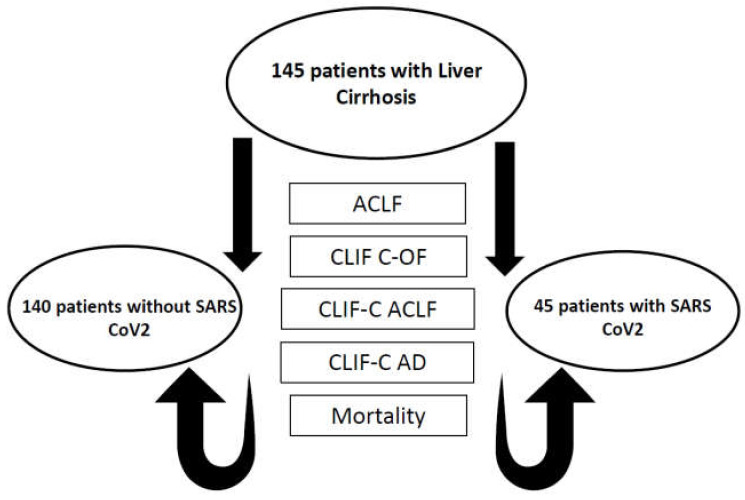
Flowchart of patient parameters evaluated in the study. AD: acute decompensation; ACLF: acute-on-chronic liver failure; CLIF-C AD: sequential organ failure assessment, acute decompensation; CLIF-C ACLF: sequential organ failure assessment, ACLF; CLIF-C OF: sequential organ failure assessment, organ failure.

**Figure 2 diagnostics-13-00600-f002:**
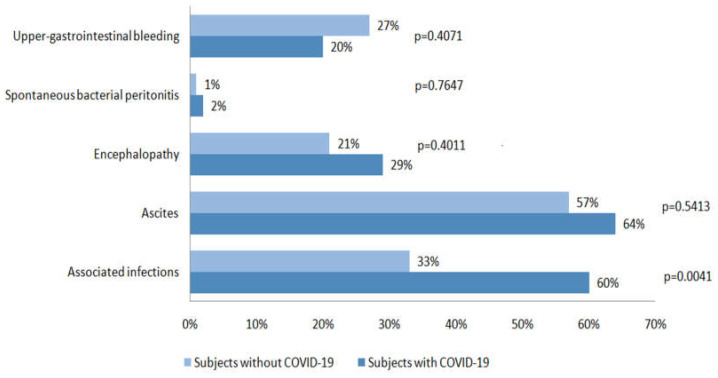
Factors of decompensation in the two groups.

**Figure 3 diagnostics-13-00600-f003:**
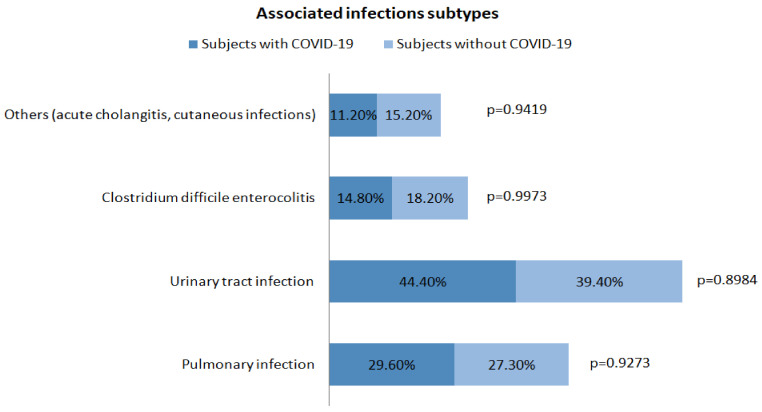
Associated infections occurring in the two groups.

**Table 1 diagnostics-13-00600-t001:** Demographic and clinical characteristics of the included subjects.

Parameter	Subjects with SARS-CoV-2 Infection *n* = 45	Subjects without SARS-CoV-2 Infection *n* = 100	*p*-Value
Age (years), mean values	62.26 ± 9.56	60.52 ± 10.55	0.3462
Gender (%)			
Female	28.9% (13)	24% (24)	0.6743
Male	71.1% (32)	76% (76)	0.7305
MELD score, mean values	20.62 ± 8.46	18.28 ± 7.59	0.0997
Liver cirrhosis etiology (%)			
ALD	57.8% (26)	56% (56)	0.9831
HCV	24.4% (11)	16% (16)	0.3311
HBV	2.2% (1)	10% (10)	0.1927
Others	15.6% (7)	18% (18)	0.9074
Child–Pugh class			
Class A	6.7% (3)	21% (21)	0.0572
Class B	28.9% (13)	33% (33)	0.7658
Class C	64.4% (29)	46% (46)	0.0613
AST (UI/L), median values	68 [19–1293]	63.5 [17–3529]	0.5011
ALT (UI/L), median values	34 [10–354]	37 [9–1669]	0.6395

MELD—model for end-stage liver disease, ALD—alcoholic liver disease, HCV—hepatitis C virus, HBV—hepatitis B virus, AST—aspartate transaminase, ALT—alanine transaminase.

**Table 2 diagnostics-13-00600-t002:** The impact of pulmonary injury on laboratory findings.

Parameter (Median Values)	Subjects with Pulmonary Injury *n* = 20	Subjects without Pulmonary Injury *n* = 25	*p*-Value
Length of hospitalization (days)	13 [1–24]	7 [1–34]	0.0159
CRP (mg/L)	56 [5.6–268]	39.4 [0.5–222.4]	0.3501
AST (IU/L)	52 [30–1293]	97.5 [19–300]	0.5093
ALT (IU/L)	30 [10–81]	47 [14–354]	0.1925
Lactate dehydrogenase (U/L)	259.5 [145–4115]	283 [243–1247]	0.4141
D-dimer (mg/L)	1201 [94–9289]	2681 [2354–4214]	0.0419
GGT	98 [19–776]	188 [29–470]	0.4182
ALP	101 [40–271]	93 [69–244]	0.8981

CRP—C-reactive protein, AST—aspartate transaminase, ALT—alanine transaminase, GGT—gamma-glutamyl transferase, ALP—alkaline phosphatase.

**Table 3 diagnostics-13-00600-t003:** Decompensation events and specific mortality.

Parameter	Subjects with SARS-CoV-2 Infection *n* = 45	Subjects without SARS-CoV-2 Infection *n* = 100	*p*-Value
No ACLF	51.1% (23)	70% (70)	0.0446
ACLF ACLF 1 ACLF 2 ACLF 3	48.9% (22) 63.6% (14) 27.3% (6) 9.1% (2)	30% (30) 76.7% (23) 23.3% (7) 0%	0.0446 0.4070 0.3509 0.1804
CLIF C-OF, median values	7 [1–12]	5 [5–10]	<0.0001
CLIF C-ACLF, median values	47.22 [29.6–60.61]	43.09 [27.7–53.51]	0.0667
CLIF C-AD, median values	54.97 [37.63–71.69]	52.83 [37.02–80.2]	0.6122
Associated infection (%)	60% (27)	33% (33)	0.0041
Overall mortality (%)	46.7% (21)	15% (15)	0.0001
Mortality in subjects without ACLF	39.1% (9/23)	12.9% (9/70)	0.0141
Mortality in subjects with ACLF	54.5% (12/22)	20% (6/30)	0.0221

ACLF—acute-on-chronic liver failure, OF—organ failure, AD—acute decompensation.

**Table 4 diagnostics-13-00600-t004:** Predictors of mortality during admission.

Baseline Parameters	Univariate Analysis		Multivariate Analysis			
			Subjects with ACLF *n* = 52		Subjects without ACLF *n* = 93	
	HR (95% CI)	*p*-Value	HR (95% CI)	*p*-Value	HR (95% CI)	*p*-Value
Pulmonary injury	1.403 (1.162–2.389)	*p* < 0.0001	1.694 (1.057–2.134)	*p* < 0.0001	1.134 (1.017–1.644)	*p* = 0.0017
The presence of COVID-19 infection	1.603 (1.362–1.935)	*p* < 0.001				
Other associated infections	1.203 (1.002–2.275)	*p* < 0.001				
Child–Pugh score	1.463 (1.042–2.015)	*p* = 0.012	1.294 (1.035–2.044)	*p* = 0.012		
MELD score	4.903 (2.002–9.275)	*p* = 0.004	1.654 (1.027–2.131)	*p* = 0.004		
ACLF grade	1.463 (1.672–2.765)	*p* = 0.017				
CLIF-OF score	7.203 (2.342–12.272)	*p* < 0.001				
CLIF C-ACLF	3.303 (1.892–5.075)	*p* = 0.003	1.914 (1.001–2.387)	*p* = 0.003		
CLIF C-AD score	1.704 (1.442–2.934)	*p* < 0.001			1.443 (1.184–1.759)	*p* = 0.0004

Cox regression analysis was used to identify factors associated with mortality during admission and used a significance level of 0.05.

## Data Availability

The data underlying the findings of the study are available on request from the corresponding author (e-mail address: alinamircea.popescu@gmail.com).

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
