# Peer review of "Impact of COVID-19 on Patients with Decompensated Liver Cirrhosis"

_diagnostics, 2023, doi:10.3390/diagnostics13040600_

Round 1
Reviewer 1 Report
This manuscript is focused on COVID-19 impact on patients with decompensated liver cirrhosis in terms of ACLF. Introduction is well written, aim is well defined and methodology section is written clearly. Results are presented with adequate tables and figures. Although this study has limitation regarding the heterogenicity of the liver cirrhosis etyology, I suggest accepting it.
Author Response
Reviewer nr 1.
This manuscript is focused on COVID-19 impact on patients with decompensated liver cirrhosis in terms of ACLF. Introduction is well written, aim is well defined and methodology section is written clearly. Results are presented with adequate tables and figures. Although this study has limitation regarding the heterogenicity of the liver cirrhosis etyology, I suggest accepting it.
Response:
Thank you for the comments. We deeply appreciate the review.
Manuscript was reviewed and English improved.

Reviewer 2 Report
The original article of Moga and co-authors is well-written and will be of great interest to the readers of the Diagnostics journal. The conclusions are consistent with the goals of this study and are supported by reliable data. The authors demonstrated that COVID-19 influenced disease progression in patients with decompensated live cirrhosis. There are some minor comments that could help to improve the manuscript.
1. It is worth expanding the discussion a bit to include published studies with animal models that will support the authors' findings.
2. The role of ACE2 overexpression in liver cirrhosis should also be clarified. Recent evidence indicates that cirrhosis significantly increases hepatic ACE2 expression (doi: 10.1136/gut.2004.062398). Pan L. reported that patients with digestive symptoms were more likely to suffer liver injury because of the upregulation of ACE-2 expression in the liver tissue (doi: 10.14309/ajg.0000000000000780). In the study of Solopov (doi.org/10.1016/j.ajpath.2022.03.012), the authors showed that mice with alcoholic liver disease had more severe acute respiratory distress syndrome compared to control mice via the activation of ACE2 receptor to the response of SARS-CoV-2 S-protein. Authors should include these papers in the manuscript.
3. Some sentences are formulated as very cumbersome and therefore difficult to understand (for example, line 247).
4. Authors should carefully re-read the entire article and correct numerous omissions and gaps in the design of the text.
Author Response
Revision of Ms: diagnostics-2178551
Dear Editor,
We deeply value the opportunity to improve our paper.
We are providing the responses to the reports of 2 reviewers. For better clarity and in order to satisfy the reviewers’ general observations, we have added information into the manuscript as indicated in the revised version, which is now being re-submitted.
We applied most of the revisions prompted by the reviewers’ comments. We hope the revised manuscript will better suit the Journal and we thank you for your continued interest in our research.
Attached are our point-by-point responses to the raised comments.
Kind regards,
Alina Popescu, MD, PhD, Proffesor
Corresponding author
Reviewer nr. 2
The original article of Moga and co-authors is well-written and will be of great interest to the readers of the Diagnostics journal. The conclusions are consistent with the goals of this study and are supported by reliable data. The authors demonstrated that COVID-19 influenced disease progression in patients with decompensated live cirrhosis. There are some minor comments that could help to improve the manuscript.
1.It is worth expanding the discussion a bit to include published studies with animal models that will support the authors' findings.
Response 1:
Thank you for the comments.
Majority of the studies including the role of ACE2 and liver fibrosis in mice are not focused on liver damage in SARS COV-2 infection and may not include population with liver cirrhosis.
However, we did find a study with very promising data but with the work currently in progress in the laboratory, questioning if ACE2 plays a role in liver fibrosis and can also have a high role in the propagation of lung injury given by SARS-CoV2 infection. (Fiona J. Warner, Harinda Rajapaksha, Nicholas Shackel, Chandana B. Herath; ACE2: from protection of liver disease to propagation of COVID-19. Clin Sci (Lond) 11 December 2020; 134 (23): 3137–3158. doi: https://doi.org/10.1042/CS20201268) “This review outlines the role of the RAS with a strong focus on ACE2-driven protective RAS in liver disease and provides therapeutic approaches to develop strategies to prevent SARS-CoV-2 infection in humans”
2.The role of ACE2 overexpression in liver cirrhosis should also be clarified. Recent evidence indicates that cirrhosis significantly increases hepatic ACE2 expression (doi: 10.1136/gut.2004.062398). Pan L. reported that patients with digestive symptoms were more likely to suffer liver injury because of the upregulation of ACE-2 expression in the liver tissue (doi: 10.14309/ajg.0000000000000780). In the study of Solopov (doi.org/10.1016/j.ajpath.2022.03.012), the authors showed that mice with alcoholic liver disease had more severe acute respiratory distress syndrome compared to control mice via the activation of ACE2 receptor to the response of SARS-CoV-2 S-protein. Authors should include these papers in the manuscript.
Response 2.
Thank you for the comments. Manuscript was reviewed and the suggestions were added.
“The complexity of the RAS (renin angiotensin system) and its modulation in liver injury is further illuminated by the overexpression of ACE2 in rat and cirrhotic human liver. ACE2 is a crucial component in the propagation of the SARS coronavirus and is a functional receptor for it. Patients with SARS frequently have liver damage, and new research has shown that the SARS coronavirus induces a unique form of hepatitis that is characterized by lobular inflammation and hepatocyte death. An explanation for the discovery that SARS infects the liver may be found in the study's conclusion that the principal cellular receptor for the disease is present in the liver and increases in liver injury [27]. Another multicntric study, which enrolled 104 patients with SARS-CoV2 infection, 81 presented digestive symptoms reported that patients with digestive symptoms were more likely to suffer liver injury because of the upregulation of ACE-2 expression in the liver tissue [28].
The study of Solopov et al. [29] focused on the role of alcohol intake and SARS-CoV 2 infection and demonstrated that in mice treated with alcohol, a histologic analysis of the lung tissue revealed aberrant immune cell recruitment in the alveolar space, abnormal parenchymal architecture, and worsened Ashcroft scores. Lung tissue homogenates from mice on an alcohol diet demonstrated upregulation of ACE2, in addition to the activation of proinflammatory biomarkers compared with mice on a control diet. This article could be very important in future research on alcoholic-induced liver disease and SARS CoV 2 infection helping in the development of therapeutic strategies. “
- Some sentences are formulated as very cumbersome and therefore difficult to understand (for example, line 247).
Response 3:
Thank you for the comments. We deeply appreciate the review.
Manuscript was reviewed and the suggestions were added.
- Authors should carefully re-read the entire article and correct numerous omissions and gaps in the design of the text.
Response 4:
Thank you for the comments. We deeply appreciate the review.
Manuscript was reviewed and the suggestions were added.
